# Fitting a lattice model with local and global transmission to spread of a plant disease

Alex Best[ID][1]*, Nik Cunniffe[ID][2]

**1** School of Mathematical and Physical Sciences, University of Sheffield, Sheffield, United Kingdom,
**2** Department of Plant Sciences, University of Cambridge, Cambridge, United Kingdom

* a.best@shef.ac.uk

## Abstract

Understanding, predicting and managing the spread of plant pathogens is crucial given the economic, societal and climatic benefits of plants, including crops and trees. Mathematical models have long been used to investigate disease dynamics in plants. An important component of such models is to account for spatial structure, since plant hosts are immobile and a majority of disease spread will often be localised. Here we apply a lattice-based mathematical modelling approach, a pair approximation, to model disease spread. While this method has previously been used to develop epidemiological theory, it has not been used to predict spread in a specific pathosystem. We fit our lattice-based epidemiological model to experimental data relating to Bahia bark scaling of citrus, an economically-important disease in north-eastern Brazil, and compare its performance to a more commonly used dispersal-kernel modelling approach. We show that the lattice-based model fits the data well, predicting a significant degree of near-neighbour infections, with similar estimated values of epidemiologically-meaningful parameters to the dispersal model. We highlight the pros and cons of the lattice-based approach and discuss how it may be used to predict disease spread and optimise control of plant diseases.

## Author summary

Plant diseases can have significant impacts, including reducing crop yields, limiting the availability of natural spaces, and the knock-on effects on our wellbeing. Mathematical models have long been used to understand how disease spreads through plant populations. Here we apply a form of mathematical model that has not previously been specifically applied to a real disease system that emphasises neighbour-to-neighbour spread of infection. In particular, we use the model to explore the spread of Bahia bark scaling of citrus, for which we have excellent experimental data available of its spatial spread. We show that the model fits

**Data availability statement:** Code is available at https://github.com/abestshef/plant_lattice_dispersal.

**Funding:** AB was funded by an EPSRC Mathematical Sciences Small Grant (UKRI1110). The funders had no role in study design, data collection and analysis, decision to publish, or preparation of the manuscript.

**Competing interests:** The authors have declared that no competing interests exist.

the data best when there is significant neighbour-to-neighbour spread with very rare long-range infections. We show that this approach agrees well with a more commonly-used mathematical framework and highlight how it might be used to test disease management strategies.

## Introduction

Diseases caused by plant pathogens can have severe consequences. Pathogens affecting crops impact yield as well as food quality and safety, leading to economic and social impacts, and in severe cases food security can even be at risk [1]. When pathogens infect plant species in natural populations, provision of ecosystem services can also be impacted [2]. Many plant pathogens are extremely well-established, causing disease on certain host species in particular locations somewhat predictably. Just two of many possible examples are late blight of potato (which remains the most significant disease of potato worldwide [3], caused by the oomycete *Phytophthora infestans*) and septoria leaf blotch of wheat (one of the most significant diseases of wheat grown in temperate climates [4], caused by the fungus *Zymoseptoria tritici*). While impacts are very difficult to unambiguously quantify, losses of yield due to these and other pathogens of major food crops are generally estimated to be in excess of 10% [5,6].

In addition to these endemic diseases, many other plant pathogens are emerging [7]. Outbreaks of emerging plant pathogens are increasingly well reported [8,9], with invasion rates appearing to increase [10,11]. Drivers are varied and complex, but are generally acknowledged to include changes to farming practices and land use [12,13], climate change [14,15], and increased travel and trade [16,17]. High profile current examples of emerging pathogens of crops include *Xylella fastidiosa* in Europe, affecting many species including olives [18] and almonds [19]; Tropical Race 4 of *Fusarium oxysporum* f.sp. *cubense*, which is spreading in many regions where banana or its wild relatives are grown [20]; *Candidatus* Liberibacter spp. causing huánglóngbìng/citrus greening, particularly in the USA and Brazil [21]; and tomato brown rugose fruit virus, which is spreading to new countries worldwide [22]. Meanwhile, in the United Kingdom alone, trees in forests are threatened by sudden larch death (caused by *Phytophthora ramorum*, the same pathogen which causes sudden oak death elsewhere [23]), chalara ash dieback (caused by *Hymenoscyphus fraxineus*, [24]) and dothistroma (red band) needle blight on conifers (caused by *Dothistroma septosporum*, [25]). Documented and ranked in the UK Plant Health Risk Register are a worryingly large number of pathogens which could potentially invade and threaten UK forests in the future [26].

Mathematical modelling is now routinely used to inform decision making around detection and control of disease outbreaks for pathogens of humans and animals [27]. However, plant disease modelling has its own long and somewhat distinctive history [28]. Models range from purely statistical, which do not attempt to capture or characterise biological mechanisms underpinning disease risk [29], to process-based compartmental models, which focus closely on interactions between members of host

population(s) with different disease statuses [30–32]. Often, for simplicity, the spatial component of disease spread is omitted, and non-spatial models are used [33]. However, since they are caused by pathogens that are actively spreading, emerging disease epidemics tend to have a particularly strong spatial component, and this requires models to include space [34]. Sometimes space is represented implicitly, focusing on pathogen spread between a small number of patches [35,36] or constituents of a metapopulation [37]. However, recent models tend to echo trends in ecology more broadly [38] by relying on the concept of a dispersal kernel, which in this context is a characterisation of the probability of inoculum from an infected host challenging a susceptible host elsewhere [34]. Early work often used a continuum description of space, either via reaction-diffusion partial differential equations [39], or integro-differential equations tracking pathogen spread [40]. Probably motivated by applications to particular pathosystems, more recent work has more often tracked disease spread in realistic geographics. Descriptions of the host population range from individual-based models [41–43] to coupled structured metapopulations, in which the disease status of the hosts in each cell on a rasterised map is tracked via its own compartmental model of disease [44,45]. Often this type of work has relied on repeated computer simulation of stochastic versions of the models, although analytic techniques are now starting to become available [46,47].

An additional subset of mathematical models for considering spatial structure in host populations are lattice-based methods [48–50]. The assumptions of these models are highly appropriate to plant systems, especially managed and agricultural populations where plants are often laid out in regular spacings. Here, individuals in a population are assumed to be fixed in a grid, with each cell occupied by at most one individual. Interactions can be assumed to be mean-field or 'global', but may also be near-neighbour or 'local'. Using a pair approximation, this system can be well described by an extended set of ordinary differential equations [48,51,52]. This deterministic model is often paired with fully spatially-explicit stochastic simulations. This approach has been used in a range of studies to explore the epidemiology, ecology and evolution of infectious disease. Early studies highlighted how local transmission leads to smaller and slower epidemics than the mean-field [53]. More recently, studies have explored how we can vary the ratio of local-to-global transmission to understand the dynamics in more detail, both at the epidemiological [54–56] and evolutionary timescale [57, 58]. However, to date these studies have almost exclusively been used to gain theoretical insights, and have not been used to directly model particular systems or be fitted to data. The closest example is that of Filipe et al. [59], who fit a purely near-neighbour lattice model to temporal experimental data of *Rhizoctonia solani* infection of radish plants, finding it fit their data well. Our work goes further by allowing the local-to-global parameter itself to be fitted to our dataset, to include limited spatial data in the fitting, and to take a Bayesian fitting procedure.

The aim of this study is to assess the utility of the lattice-based approach in understanding disease spread in real-world systems. In particular, we will focus on previously published data set on Bahia bark scaling of citrus (BBSC), a disease of unknown etiology which has been endemic in north-eastern Brazil since the 1960s [60]. This system is a useful comparison since (a) the experimental study was of citrus trees laid out in a simple grid pattern, matching the assumptions of the model well, and (b) a study fitting an individual-based model representing transmission via a dispersal kernel to the data has already been undertaken [61]. They showed their model could fit the BBSC data well, giving us a good comparison for our parameter estimates.

## Materials and methods

### Experimental data

We use published experimental data for BBSC to fit our models [60]. In the experiment underlying this work, a $15 \times 16$ grid of initially immature citrus trees was established with a 2m x 2m spacing adjacent to a small stand of older trees showing BBSC symptoms. These established founder trees were not in the same spatial arrangement as the experimental grove, with greater distances between each individual, as well as between the established trees and experimental grove. As mentioned below, this initial arrangement of founder trees in the experiment needed to be expressed differently in the model. In particular, we assume (in both models) that there is a single line of infected individuals along the

southern edge of the tree stand, since we require equal spacing for the lattice model assumptions to hold. The first infection was detected around 30 months after the experimental grove was planted. Over a period of 5 years, the trees were then inspected 14 times and those showing signs of infection recorded. We therefore know both the total number of trees symptomatically infected at each sample point as well as their spatial location. Snapshots of the experimental data are shown in supplementary S1 Fig.

### The epidemiological model

Our model follows a Susceptible - Exposed - Infected compartmental framework, with no births, deaths or recovery. This is in line with the previous modelling study of BBSC [61] and with the nature of the experimental system [60]. Let $P_S$, $P_E$, $P_I$ and $P_F$ be the probabilities that, if we choose a random site on the lattice, it is occupied by, respectively, a susceptible, exposed, infected or 'founder'. The founder trees, $P_F$ are those trees initially inoculated at the start of the experiment, one row outside of the main experimental tree stand. Since there is no recovery, $P_F$ is therefore a constant. Further, as there are no empty sites on the lattice, $P_S + P_E + P_I + P_F = 1$ always. As such, we need only track two of these variables with differential equations. Also let the term $q_{I/S}$ be the conditional probability that if we choose a susceptible individual, it has a neighbour that is infected, and similarly for $q_{F/S}$. We can describe the dynamics of the 'singlet' densities by the following equations,

$$\frac{dP_S}{dt} = -\beta \left( L q_{I/S} + (1-L) P_I \right) P_S - \beta_F \left( L q_{F/S} + (1-L) P_F \right) P_S, \tag{1}$$

$$\frac{dP_E}{dt} = \beta \left( L q_{I/S} + (1-L) P_I \right) P_S + \beta_F \left( L q_{F/S} + (1-L) P_F \right) P_S - \rho P_E. \tag{2}$$

The parameter $\beta$ gives the rate of infection from infected trees, and $\beta_F$ that from founder trees. We allow the infection rates from the two types of tree to differ for a number of reasons: the founder trees were much larger, they had been infected for some time at the start of the experiment, and the spatial distribution of founder trees in the model is slightly different to that of the experiment (described in detail above). The parameter $\rho$ is the rate of progression from being exposed to being infectious (with $1/\rho$ giving the latent period). The parameter $L \in [0, 1]$ controls the ratio of local-to-global infection. We stress here that 'local' infections are purely near-neighbour, and 'global' infections are random/mean-field, such that $L = 0.5$ means that half of an individual's infectious contacts are from neighbouring sites and half from anywhere else on the lattice. If $L = 1$ the Eqs (1)–(2) are simply the standard mean-field SEI model. Note we assume this is the same for all trees - we felt justified in this assumption given initial exploratory work where we included a separate parameter, $L_F$, for founder trees identified that the posterior distributions of the parameters $L$ and $L_F$ were almost identical.

This system is not closed, since, for example, $q_{I/S} = P_{SI}/P_S$, where $P_{SI}$ is the probability that if we choose a neighbouring pair of sites, one is susceptible and the other infected. We therefore need to write down equations describing the dynamics of these 'pair' densities. We can find these by considering the possible transitions between pairs of sites. For example, consider the creation and loss of an [S,I]-pair:

- $[S, E] \rightarrow [S, I]$ (created) $E$ member of an [S,E]-pair becomes symptomatic at rate $\rho$
- $[S, I] \rightarrow [E, I]$ (lost) $S$ member of the pair becomes exposed after infection from
  - its known $I$ neighbour (i.e. from the [S,I]-pair) at rate $\beta L/4$
  - another $I$ neighbour (since each individual has 4 neighbours) at rate $3\beta L q_{I/SI}/4$
  - a random $I$ contact at rate $\beta(1-L)P_I$
  - a $F$ neighbour at rate $3\beta_F L q_{F/SI}/4$
  - a random $F$ contact at rate $\beta_F(1-L)P_F$.

Considering all such possible transitions we reach the following set of equations,

$$\frac{dP_{SS}}{dt} = -2\left[\beta\left(\frac{3}{4}Lq_{I/SS} + (1-L)P_I\right)P_{SS} + \beta_F\left(\frac{3}{4}Lq_{F/SS} + (1-L)P_F\right)P_{SS}\right], \tag{3}$$

$$\frac{dP_{SE}}{dt} = -\beta\left(\frac{3}{4}Lq_{I/SE} + (1-L)P_I\right)P_{SE} - \beta_F\left(\frac{3}{4}Lq_{F/SE} + (1-L)P_F\right)P_{SE} - \rho P_{SE}$$
$$+ \beta\left(\frac{3}{4}Lq_{I/SS} + (1-L)P_I\right)P_{SS} + \beta_F\left(\frac{3}{4}Lq_{F/SS} + (1-L)P_F\right)P_{SS}, \tag{4}$$

$$\frac{dP_{SI}}{dt} = -\beta\left(L\left[\frac{1}{4} + \frac{3}{4}q_{I/SI}\right] + (1-L)P_I\right)P_{SI} - \beta_F\left(\frac{3}{4}Lq_{F/SI} + (1-L)P_F\right)P_{SI}$$
$$+ \rho P_{SE}, \tag{5}$$

$$\frac{dP_{EE}}{dt} = -2\rho P_{EE} + 2\beta\left(\frac{3}{4}Lq_{I/SE} + (1-L)P_I\right)P_{SE}$$
$$+ 2\beta_F\left(\frac{3}{4}Lq_{F/SE} + (1-L)P_F\right)P_{SE}, \tag{6}$$

$$\frac{dP_{EI}}{dt} = -\rho P_{EI} + \beta\left(L\left[\frac{1}{4} + \frac{3}{4}q_{I/SI}\right] + (1-L)P_I\right)P_{SI} + \beta_F\left(\frac{3}{4}Lq_{F/SI} + (1-L)P_F\right)P_{SI}$$
$$+ \rho P_{EE}, \tag{7}$$

$$\frac{dP_{II}}{dt} = 2\rho P_{EI}, \tag{8}$$

$$\frac{dP_{SF}}{dt} = -\beta\left(L\frac{3}{4}q_{I/SF} + (1-L)P_I\right)P_{SF} - \beta_F\left(L_F\left[\frac{1}{4} + \frac{3}{4}q_{F/SF}\right] + (1-L_F)P_F\right)P_{SF}, \tag{9}$$

with $P_{SS} - P_{II} - P_{EE} - P_{FF} + 2(P_{SE} + P_{SI} + P_{EI} + P_{EF} + P_{IF}) = 1$. This system is also not closed, since we have conditional probabilities such as $q_{I/SS} = P_{SSI}/P_{SS}$, which would require writing down the dynamics of 'triplet' densities. At this point we can apply a pair approximation, letting $q_{I/SS} = q_{I/S}$, which closes the system [48]. While this model comprises a large set of equations, it only includes four parameters: the infection parameters from infected ($\beta$) and founder ($\beta_F$) trees, the emergence of infectivity rate, $\rho$, and the parameter controlling the ratios of local-to-global transmission, $L$.

We also produce stochastic simulations of the system using a direct-method ('Gillespie') algorithm [62]. This uses the singlet equations from the lattice model (Eqs (1)–(2)) to define the event rates, but unlike the deterministic model it is fully spatially explicit, with dispersal either near-neighbour (with probability $L$) or random (with probability 1–$L$). While not used in the model fitting, results from this model allow us to check that the output of fully spatially-explicit simulations match with the predictions of the deterministic but approximate model. These simulations are also required for looking at detailed spatial questions, such as the level of infection on certain rows of the grid.

For comparison, we also produce a dispersal model. This has the same underlying epidemiological assumptions, but now rather than the simple local/global delineation, the way in which the probability of infection decays with increasing distance is parameterised using a dispersal kernel. In keeping with the previous study [61], we assume the dispersal kernel is exponential. We calculate the distance, $d_{ij}$ between two trees $i$ and $j$ (treated as the central point of the grid cells for direct comparison to the lattice-based pairs model), and then take the dispersal kernel between a susceptible individual $S_i$ and infected individual $I_j$ to be,

$$\phi_{ij} = \Lambda\frac{\exp(-d_{ij}/\alpha)}{2\pi\alpha^2},$$

where $\alpha$ is the scale parameter (and the corresponding mean dispersal distance in a homogeneous landscape is $2\alpha$ [34]). In our model one 'unit' of distance is the space between two trees, which in the experiment was 2m. Therefore the 'real-world'-adjusted mean dispersal distance would be $4\alpha$. The parameter $\Lambda$ accounts for the typical (crown) area of trees, and

makes the dispersal kernel a dimensionless quantity. Therefore the transmission rate experienced by a susceptible individual, $S_i$, is, $\beta \sum_j \phi_{ij} I_j$. As described in [61], we arbitrarily set $\Lambda = 1\text{m}^2$, noting that any value could have been used since any choice can be scaled into the fitted infection rate parameters, $\beta$ and $\beta_F$.

Note that this system cannot be analysed deterministically. As such we produce an additional direct-method stochastic algorithm for this system. The parameters for this model are again infection rate parameters, $\beta$ and $\beta_F$, progression rate $\rho$ and now dispersal scale parameter, $\alpha$.

As in the previous study that focussed on a dispersal model for this data [61], we also include an initial time delay in both models to account for the time it takes for the (non-founder) plants within the grove to reach epidemiological maturity, and so the delay before the first infections can occur, $\delta$. Since the first infections were detected 30 months after the beginning of the experiment, we assume $0 \leq \delta \leq 40$ in our model fitting.

## Model fitting

We use an Approximate Bayesian Computation (ABC) approach to fit the models to the data. In a Bayesian framework, the parameter values are treated as unknown random variables. Our aim is to calculate the likelihood that certain parameter values ($\theta$) would generate the data ($D$). In particular, we can use prior information we might have about the distribution of the parameter values, $\pi(\theta)$, in combination with the likelihood of the data, $\pi(D|\theta)$, to arrive at posterior distributions for the parameter values,

$$\pi(\theta|D) = \frac{\pi(D|\theta)\pi(\theta)}{\int_\theta \pi(D|\theta)\pi(\theta)d\theta}$$

A considerable difficulty is computing the likelihood, $\pi(D|\theta)$. The ABC approach negates this issue by simulating the model and considering how closely model outputs (for example the number of infected plants) match the recorded data points. The key to this approach is, rather than looking for a single 'best fit', we keep all parameter combinations where the fit is 'good enough', and use the proportion of simulations for a given parameterisation that meet this constraint as a proxy for the likelihood function. ABC approaches are increasingly applied to a range of biological models including epidemic models [63–67].

We take a relatively straightforward rejection-ABC approach, using the experimental data described above. For our fitting function, we take the sum of squared error between the data (expressed as proportions) at each of the recorded timepoints and the model prediction for the number of infections ($P_I$) and the infected-infected pair densities ($P_{II}$) at these timepoints. Since the experimental data only records the number of infections in the experimental tree stand, we re-calculate $P_I$ and $P_{II}$ to be the relative proportion of *non-founder* trees. To be accepted, the errors of both $P_I$ and $P_{II}$ must be within some threshold. Clearly, the posteriors will vary depending on the chosen acceptance threshold. We present results in the main text for a single chosen threshold, but show results for other values in the supplementary information (S4 and S5 Figs) and comment on our choice of default acceptance threshold. We use uniform prior distributions for all parameters, with $L \in [0, 1]$ (dimensionless), $\beta, \beta_F \in [0, 5]$ months$^{-1}$, $1/\rho \in [0, 20]$ months (note we are using the latent period rather than the direct parameter $\rho$ here as it has clearer biological significance), and $\delta \in [0, 40]$ months. In the dispersal model we take $\alpha \in [0, 10]$ ($\alpha$ has units of distance, with $2\alpha$ the mean dispersal distance, but in the model one 'unit' of distance equates to 2 metres in the experiment).

For both the lattice model and dispersal model we run 5 million simulations on a High Performance Computing cluster. All simulations are conducted in Python. The lattice model is run using solve_ivp() with the Radau solver from the SciPy package [68]. Python code to run the models is available at

`https://github.com/abestshef/plant_lattice_dispersal`.

# Results

## Lattice model

We begin by examining the posterior distributions for the lattice model. We choose an error acceptance threshold of 0.025 for the results presented in the main text, for which 598 (0.012% of the 5 million runs) parameter sets were accepted. In the supplementary S4 Fig we show posterior distributions for a number of different thresholds for comparison. We find that our chosen threshold has enough accepted fits to form 'smooth' posteriors, and the distributions remain consistent for small changes to the threshold.

The posterior distributions are plotted in Fig 1 and the mode for each parameter (after smoothing from a Gaussian kernel density estimator) and its 90% Smallest Credible Interval (sometimes known as Highest Density Interval) are shown in Table 1. The local-to-global ratio parameter, $L$, shows a significant skew, with a mode of 0.99, and no values of $L < 0.70$ accepted, indicating that predominantly localised spread is likely. For the two transmission parameters - $\beta_F$ for infections from founders and $\beta$ for all secondary infections - the distributions are notably different, with a much broader distribution for the founder infections. The distribution for the latent period is reasonably symmetric around a mode of 10.53 months, while $\delta$ is left-skewed with a mode of 27.57 months. We show correlations between the parameters in supplementary S2 Fig, which highlight that $\beta$ and $1/\rho$ have a strong non-linear correlation, with a much broader spread of transmission rates for high latent periods. We also see that low $L$ values require low $\beta$ values.

We conduct a posterior predictive check of the dynamics in Fig 2 by randomly choosing 100 sets of parameter values from our joint posterior distributions and plotting the dynamics against the data. We see remarkably good fits visually from the model to both data sets (single and pair densities).

We also used the posterior distributions from the deterministic model to run fully spatially-explicit stochastic simulations of the model to check that the results are similar to the approximate deterministic model. Fig 3A reveals that while the visual fit is still good, the stochastic simulations do show more variation than the deterministic model; this is to be

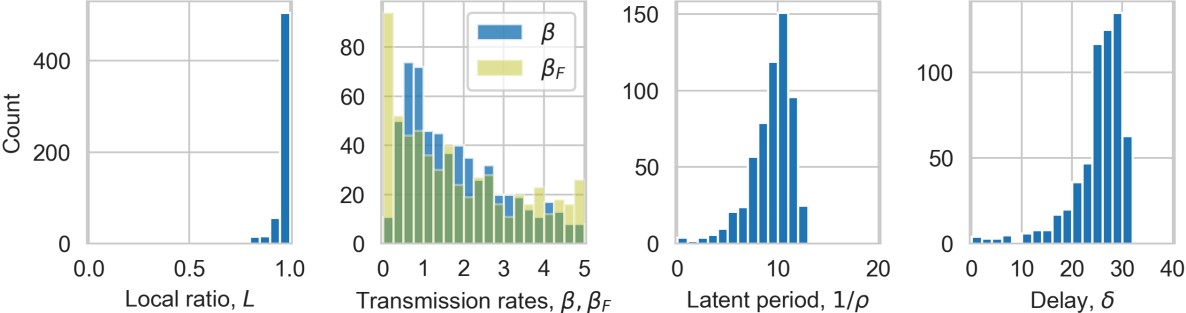

**Fig 1**. **Posterior distributions for the lattice model.** Histograms of the number of simulation accepted by the ABC procedure for local-to-global ratio, $L$, transmission parameters for regular, $\beta$, and founder $\beta_F$ infections, latent period $1/\rho$ and delay, $\delta$.

**Table 1**. **Mode parameter values for the lattice model (using a Gaussian kernel density estimator) and the 90% Smallest Credible Interval/Highest Density Interval from the posterior distributions with an error threshold of 0.025.**

| Parameter | Mode | 90% Smallest Credible Interval |
|---|---|---|
| $L$ | 0.99 | [0.93, 1.00] |
| $\beta$ (months$^{-1}$) | 0.82 | [0.18, 3.70] |
| $\beta_F$ (months$^{-1}$) | 0.48 | [0.01, 4.26] |
| $1/\rho$ (months) | 10.53 | [6.58, 12.40] |
| $\delta$ (months) | 27.57 | [18.66, 31.87] |

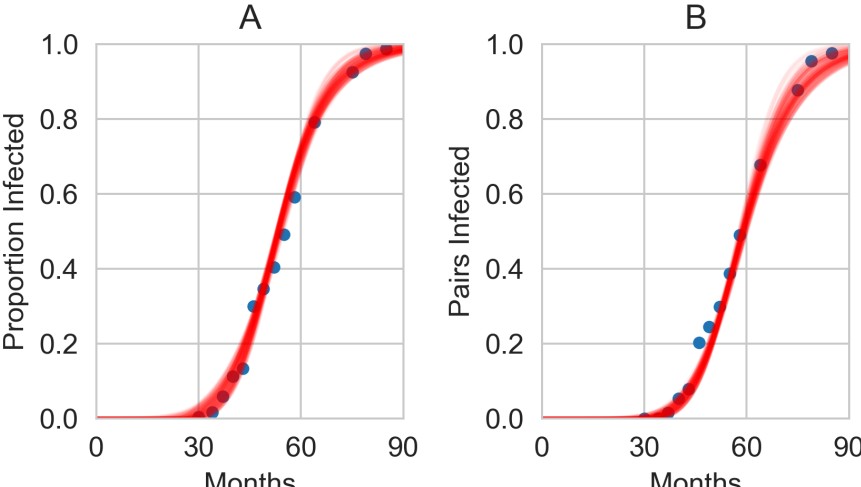

**Fig 2. Posterior predictive check of the lattice model.** 100 simulation runs of the lattice model (Eqs (1)–(9)) using parameters randomly drawn from the joint posterior distributions in Fig 1 shown in red. The blue dots mark the data points from the BBSC experiments. Panel (a) shows the total proportion infected, panel (b) the proportion of Infected-Infected neighbouring pairs.

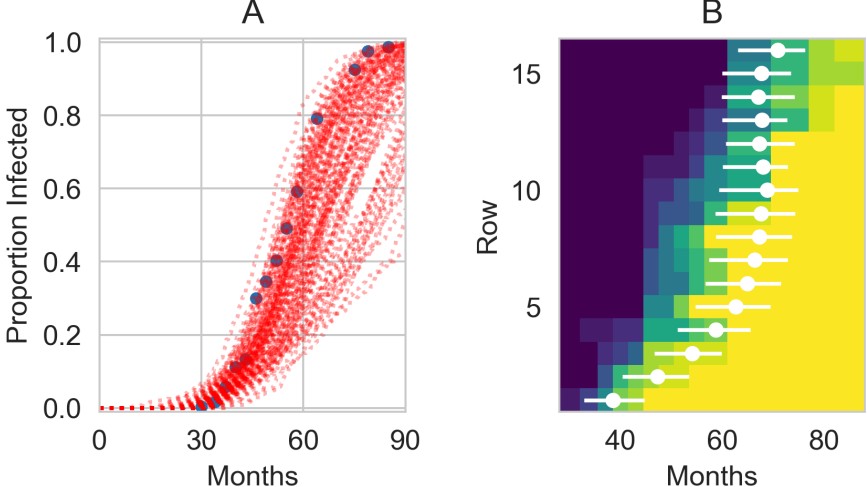

**Fig 3. Stochastic realisations of the lattice-fitted model using the joint posterior distributions.** (a) Time-courses from 100 stochastic simulation runs alongside the data (blue circles). (b) The month when infection arrives on each row. The colormap shows the proportion of trees infected from the data at each time-point, with blue colours meaning low prevalence and yellow colours high prevalence. The white dots are the the average of the 100 simulations for the median (8th) tree on the row being infected and the lines for the 4th and 11th trees.

expected due to the inherent stochastic nature of these simulations, but may potentially also be due to the more detailed spatial organisation included in the simulations. We can also use these simulations to examine the spatial spread of the infection by considering how quickly it spreads to each row (Fig 3B), as was similarly done in the previous modelling study of these experimental data [61]. We see that across the first 6-8 rows there is a trend of increasing time for the infection to reach the row that is roughly in line with the data. Past this point, the model predicts a fairly steady arrival time, due to the effect from the lattice model of infection being either near-neighbour or random; the rows at a distance from the source are almost equally likely to be infected through random infection.

As a guide, we also produce a series of snapshots of the grid for one randomly chosen stochastic realisation of the lattice model using parameters from the joint posterior distribution in Fig 4, with all parameters within the 90% smallest credible intervals reported in Table 1. While the expansion is primarily localised, we see occasional longer-range infections which seed new disease foci.

## Dispersal model

We now examine the dispersal model to which we compare our lattice model. We again show results in the main text for a threshold of 0.025 for consistency, and show the posteriors for other thresholds in the supplementary S5 Fig. We note that we do see slightly higher rates of acceptance for this model (3849 accepted sets, which is 0.08% of the 5 million runs), but caution is needed when comparing the acceptance rates with those of the lattice model given the differing natures of the parameters $L$ and $\alpha$.

The posterior distributions are shown in Fig 5 and the modes and 90% smallest credibile intervals in Table 2. Similarly to the lattice model, there is clear skew towards more localised infection, with mode $\alpha = 0.74$ (giving a 'real-world' mean dispersal distance of 2.96m), and only 26 parameter sets were below the acceptance threshold where $\alpha > 5$. The distributions for the transmission parameters are similar to the lattice model, with a much broader distribution for $\beta_F$ than for $\beta$. Again, similarly to the lattice model, we see a symmetric distribution for $\rho$ with a mode of 8.19 months, and a slightly left-skewed distribution for $\delta$ (mode 30.00 months). We again show correlations between the parameters in supplementary S3 Fig, which show similar correlations between $\beta$, $1/\rho$ and $\alpha$, although with much more noise than for the lattice model.

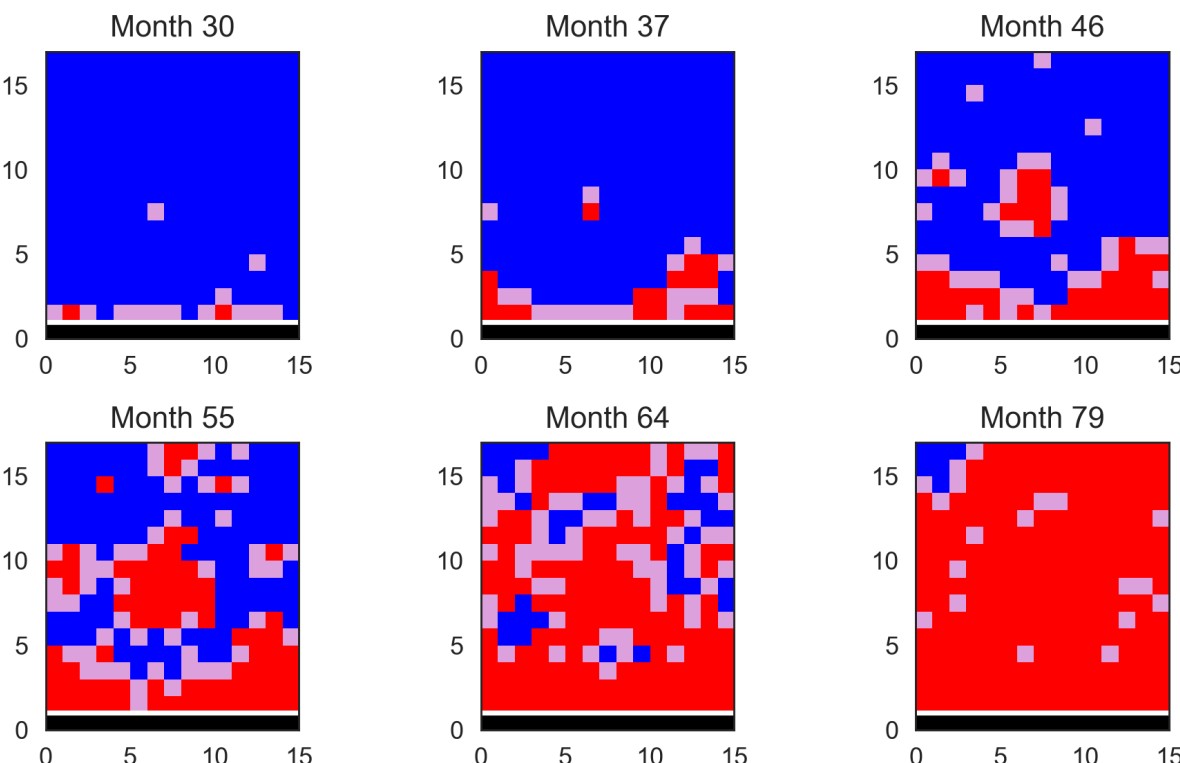

**Fig 4**. **Snapshots of the grid showing the spatial spread of the disease in one stochastic realisation of the lattice-fitted model.** Blue denotes a cell with a susceptible plant, purple a cell with an exposed plant and red a cell with an infected plant, with the row of founder trees in black. In this simulation run, $L = 0.98$, $\beta = 1.21$ months$^{-1}$, $\beta_F = 0.69$ months$^{-1}$, $1/\rho = 9.79$ months and $\delta = 22.71$ months.

PLOS Computational Biology

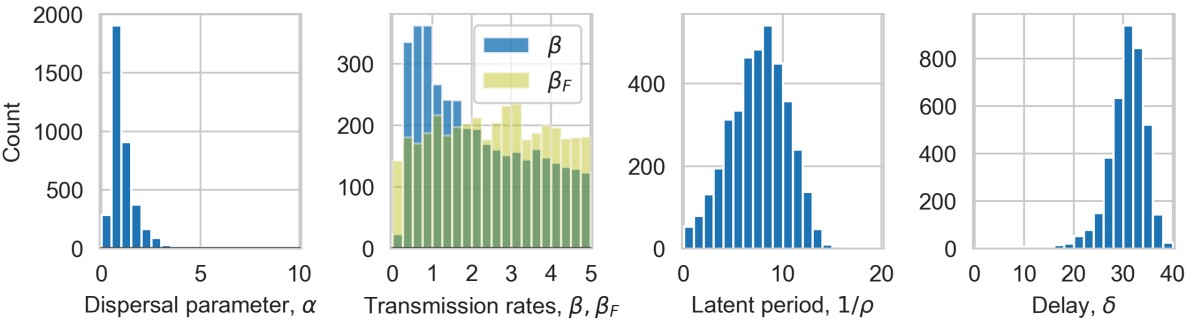

**Fig 5**. **Posterior distributions for parameters from the dispersal model.** Histograms of the number of simulation runs kept by the ABC procedure for dispersal ($\alpha$), regular ($\beta$) and founder ($\beta_F$) transmission, latent period $1/\rho$ and delay, $\delta$.

**Table 2**. **Mode parameter values for the dispersal model (using a Gaussian kernel density estimator) and the 90% Smallest Credible Interval/Highest Density Interval from the posterior distributions with an error threshold of 0.025.**

| Parameter | Mode | 90% Smallest Credible Interval |
|---|---|---|
| $\alpha$ | 0.74 | [0.35, 2.02] |
| $\beta$ (months$^{-1}$) | 0.74 | [0.25, 4.29] |
| $\beta_F$ (months$^{-1}$) | 2.93 | [0.39, 4.78] |
| $1/\rho$ (months) | 8.19 | [2.82, 12.27] |
| $\delta$ (months) | 30.00 | [25.24, 36.51] |

Conducting the posterior predictive checks we see more variation than in the deterministic lattice model (Fig 6A). As might be expected, they are visually more similar to the stochastic simulations of the lattice model (Fig 3A). This is again due to the fact that even using the posterior distributions, the stochastic nature of the simulations means that we can have very different time-courses for the same parameter sets.

We also compare the times for the infection to spread to each row (Fig 6B). We see a clear trend of increasing times to reach each row, and the dispersal model visually matches the data better than the equivalent plots for the lattice model (Fig 3B). For the snapshots of the grid from one stochastic realisation of the dispersal model (Fig 7), with parameters chosen from within the 90% credible intervals in Table 2 we see a steady, though fragmented, spread of the infection through the rows.

## Comparison of run times

An important point to note when considering the comparison of the lattice and dispersal models is their run times. Since the lattice model is deterministic, it runs considerably faster than the stochastic dispersal model. As an example, the run-time for a batch of 100,000 simulations of the lattice model on an HPC node was around 50 minutes, but for the dispersal model was over 60 hours, more than 70 times slower. While the particular run times are not important, and while efficiency savings might be made (for example by coding in Fortran or C rather than Python), the important conclusion is that based on this result we would certainly expect the lattice model to be at least an order of magnitude faster than the stochastic dispersal model. We note that this runtime for the dispersal model will only increase with larger populations, but should stay near constant for the lattice model since, as a deterministic ordinary differential equation model, there is no dependence on population size.

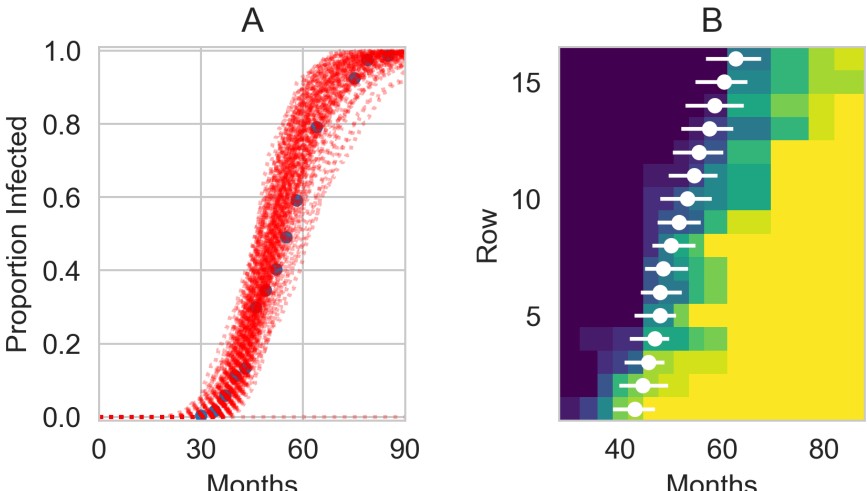

**Fig 6**. **Stochastic implementations of the dispersal model using the joint posterior distributions.** Left: time-courses from 100 stochastic simulation runs using parameter values selected from the joint posterior distributions alongside the data (blue circles). Right: the month when infection arrives on each row. The colormap shows the proportion of trees infected from the data at each time-point, with blue colours meaning low prevalence and yellow colours high prevalence. The white dots are the the average of the 100 simulations for the median (8th) tree on the row being infected and the lines for the 4th and 11th trees.

## Discussion

We have demonstrated that a lattice-based, deterministic epidemiological model with a mix of local and global infection can provide a good fit to experimental data concerning the spread of a plant pathogen. In particular, an experimental study tracking spread of Bahia bark scaling of citrus (BBSC) through a grid-like stand of fruit trees [60] is well approximated by a model with predominantly local transmission, but including occasional global (i.e., longer distance) dispersal. In broad terms, and although the wave-like spread of the disease front in space was not fully captured by the lattice model, its performance was similar to that of a dispersal kernel model (a fully spatially explicit individual based model) as used previously both for this pathosystem [61] and a number of others (e.g. citrus canker [41,42,69,70], citrus greening/huanglongbing [43,71,72], citrus tristeza virus [73], sudden oak death [74,75], cassava brown streak [76] and *Xylella fastidiosa* [77]). For BBSC, both the lattice-based and dispersal kernel models predict highly localised spread with modal parameter values of 99% local interactions / 2.9m dispersal distance, a latent/asymptomatic period of 8-11 months and a maturation delay of 27-30 months. Our 90% smallest credible intervals reliably overlay the reported 95% confidence intervals of the previous model based on the dispersal kernel formulation fitted to these data by Cunniffe et al. [61] .

It might well be questioned how realistic the underlying assumption of the lattice model is, i.e., that infection is *either* local *or* global. Of course, in many real-world systems, the more granular dispersal kernel is likely to be a more realistic interpretation of disease spread. However, we might propose situations where this assumption makes some sense, at least as an initial approximation. For example, spores of *Dothistroma septosporum*, a commercially important disease of pine and fir trees, have two morphs - asexually reproduced conidia that are primarily splash-dispersed over short distances and sexually reproduced ascospores which are wind dispersed over much greater distances [78]; *Hymenoscyphus fraxineus*, the fungal pathogen causing ash dieback, also disperses in this fashion [24]. Similar separations of dispersal scale caused by different types of propagules are a feature of the epidemiology of a number of other diseases caused by ascomycete pathogens of crops, including septoria on wheat [4], powdery mildew on a number of host species [79], and blast diseases of wheat [80] and rice [81]. Although it would also involve a separation of timescales, a somewhat similar

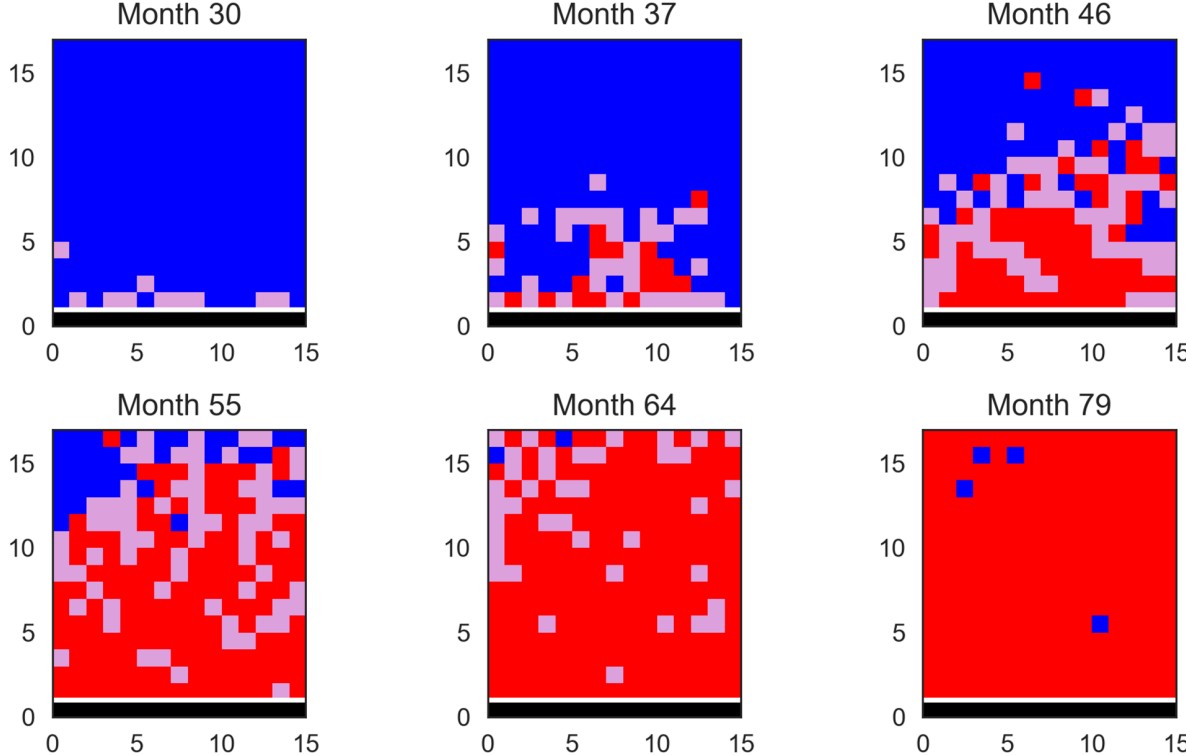

**Fig 7. Snapshots of the grid showing the spatial spread of the disease in one stochastic realisation of the dispersal model.** Blue denotes a cell with a susceptible plant, pink a cell with an exposed plant and red a cell with an infected plant, with the row of founder trees in black. In this simulation run, $\alpha = 0.88$, $\beta = 1.61$, $\beta_F = 0.85$, $1/\rho = 7.65$ and $\delta = 27.20$.

mechanism could partially underpin soil-borne pathogen spread patterns, with spread often very restricted within a growing season, but with movement over long distances by cultivation at harvest time [82]. We might also consider vector-borne diseases where there is dimorphism in the vector between winged and non-winged forms. Aphids, for example, have morphs who can only crawl between neighbouring plants and those who can fly long distances [83], likely leading to a mix of more local and more global dispersal. This would be particularly apparent for diseases which are vectored by a mix of colonising aphids and transient aphids [84], a distinction based on whether the aphid population is resident on plants in a particular field or instead only briefly pass through probing a small number of plants essentially at random before flying off elsewhere. Beyond infectious diseases, our model framework would also be relevant for insect pests such as oak processionary moth, where caterpillars will tend to only spread short distances but moths much farther [85].

Perhaps the strongest reason for using the lattice model over the dispersal model is a practical one. Since the lattice-based approach yields a deterministic model, the Monte Carlo fitting procedure, with many thousands of model runs needed to form meaningful posterior distributions, is orders of magnitude quicker compared to a stochastic implementation. While more advanced computational techniques would improve the speed - utilising GPUs for example - this time difference between the approaches will always remain to some extent. This difference in computational speed will only grow as we look at larger populations of plants. The speed of the deterministic model will remain almost unchanged at higher population sizes, but the stochastic implementations will become significantly slower and may start needing their own approximations, such as tau-leaping, to become practical to investigate [86]. We would thus propose the lattice model as, at the least, an initial choice for investigating disease spread in larger plant populations. Indeed, given the similarity in

posterior distributions between the two approaches found here, we might find it useful to take a sequential approach, fitting the lattice model first to find parameter distributions that could then be taken forward to be used as priors in a dispersal model.

While the speed of the model fitting will be considerably faster for the determinstic lattice model in a larger population, the assumption of mean-field transmission will likely become less reasonable. A possible approach here would be to extend the framework of Ellner [50] for multiple interaction scales in a pair approximation model. In the original study, this was applied such that two different ecological processes - competition and infection, say - could be considered to act over different distances. However, we could equally use this approach to define two or more transmission scales, in some sense giving a discretised dispersal kernel. Here we used a simple von Neumann neighbourhood of four near-neighbours on a square lattice, but alternative neighbourhood definitions could also be used to alter the scale of local infection.

The dispersal kernel model used here for comparison with our lattice-based model was inherited from the original study of Cunniffe et al. [61], and used an exponential dispersal kernel. The probability of transmission decays exponentially with distance, a so-called thin-tailed kernel. Fat-tailed kernels - which decay more slowly than exponentially in their tails - are often supported by empirical plant disease data (see Fabre et al. [34] for a review). Such kernels are well known to have important epidemiological consequences, most notably permitting disease wavefronts that accelerate over time [87–89]. The choice of a thin-tailed kernel in [61] was motivated by apparently highly localised transmission and a relatively constant wave speed in the experimental data, and strong support from exploratory analyses comparing alternative dispersal kernel formulations. In this context, it is therefore striking that we were able to recover a good fit to the experimental data using the local–global model, since the equivalent dispersal structure is in fact closer to a fat-tailed kernel, since the global component of dispersal corresponds to an equal probability of dispersal between any pair of trees, irrespective of distance. However, perhaps as a consequence of the limited spatial extent of the experimental plot, this did not result in substantial systematic lack of fit.

We have used this initial study as a 'proof of concept' of using lattice-based models to study disease spread in plant populations. Only one previous study had used a lattice-based model to fit to experimental data [59], which assumed entirely near-neighbour infection, and also showed good agreement between the model and data. There are a number of developments that might be made in future work. We may well wish to explore more efficient and/or detailed statistical methods for fitting the models. The ABC rejection approach used here is relatively simple to implement but can have its drawbacks [90]. As already alluded to, the fitting procedure could be improved with a sequential approach - undertaking an initial run to determine the broad shapes of the distributions, and then subsequent runs to explore these in more detail. Recent research has also explored using machine learning to improve the ABC algorithm [91]. It would also be possible to include more detail on the spatial pattern of disease in the pseudo-likelihood used in the ABC procedure, for example by going beyond simple statistics on numbers/pairs of infected/susceptible hosts, and instead including information on the full distribution of distances between infected plants on successive surveys [66], or the Earth Mover's Distance [92] between the simulation runs and the spatial data. Indeed, the Markov Chain Monte Carlo inference algorithm presented in the original study of [61] used information on the location of each host, as well as its symptom status over time, in order to calculate the full likelihood of the data given model parameters. However, doing so relied on data augmentation, an approach dating back to the pioneering work of [73] (in fact first used in the context of epidemiology for citrus tristeza virus, another citrus disease). Briefly, the approach allows unobserved and/or censored determinants of the model's likelihood function - here the unobservable time of first infection of each host, and the censored time of symptom emergence - to be treated as additional unknown parameters to be estimated within ranges consistent with the constraints from the data, thereby allowing likelihood-based inference to be performed. While MCMC with data augmentation has subsequently been used successfully for a number of plant disease systems (see, e.g., [69–71,93]), it tends to be most useful for systems consisting of a relatively small number of individuals, since the number of additional parameters to estimate scales roughly linearly with the system size. ABC approaches are therefore becoming increasingly common for fitting models at larger spatial scales [30].

More practically, an important development will be to use the models to explore prevention and mitigation strategies. For example, roguing, removing (and potentially replacing) symptomatic plants, is commonly used as a control strategy [94]. The effects of this have been explored by a number of mathematical models [45,61,72,95–101] and could easily be included in our lattice model. Moreover, it would be relatively straightforward to include a strategy of removing plants that neighbour symptomatic cases in our model set-up. It is clear from the snapshots that, given the incubation period of 8-11 months in both models, there are often a large number of asymptomatic trees present at any time, and only removing symptomatic trees is unlikely to fully control the epidemic [34,42,44,70,75,77,102,103]. Exploring these management strategies is an important tool offered by mathematical models, and can lead to evidence-based strategies for disease control in real populations.

## Supporting information

**S1 Fig. Visualisation of experimental data.** Red squares denote symptomatic trees and blue squares non-symptomatic, with the row of founder trees in black. Note that unlike the model snapshots it is not possible to detect trees in their latent infected period.
(TIFF)

**S2 Fig. Correlations between the posterior parameters from the lattice model.**
(TIFF)

**S3 Fig. Correlations between the posterior parameters from the dispersal model.**
(TIFF)

**S4 Fig. Posterior density plots for the lattice model for different error acceptance thresholds.** The default value of 0.025 is shown in bold, with a range of higher and lower values also presented. The distributions for each parameter appear quite consistent and, as is expected, there is a clear narrowing of the distributions as the threshold is reduced. The distributions appear to temporarily stabilise between values of 0.03 and 0.0225. While there is then some further movement for the lower threshold in some parameters, the number of acceptances here is rather low and we felt it a less reliable guide.
(TIFF)

**S5 Fig. Posterior density plots for the dispersal model for different error acceptance thresholds.** The default value of 0.025 is shown in bold, with a range of higher and lower values also presented. The distributions for each parameter appear quite consistent and, as is expected, there is a clear narrowing of the distributions as the threshold is reduced. The distributions again stabilise between values of 0.03 and 0.0225.
(TIFF)

## Acknowledgments

Thanks to Chico Laranjeira for making the BBSC data available via their 2006 publication.

## Author contributions

**Conceptualization:** Alex Best.

**Formal analysis:** Alex Best.

**Investigation:** Alex Best.

**Methodology:** Alex Best, Nik Cunniffe.

**Project administration:** Alex Best.

**Resources:** Alex Best.

**Software:** Alex Best.

**Validation:** Alex Best, Nik Cunniffe.

**Visualization:** Alex Best.

**Writing – original draft:** Alex Best, Nik Cunniffe.

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
