## [Decision Letter · Decision Letter 0]

14 Nov 2025

PCOMPBIOL-D-25-01589

Fitting a lattice model with local and global transmission to spread of a plant disease

PLOS Computational Biology

Dear Dr. Best,

Thank you for submitting your manuscript to PLOS Computational Biology. After careful consideration, we feel that it has merit but does not fully meet PLOS Computational Biology's publication criteria as it currently stands. Therefore, we invite you to submit a revised version of the manuscript that addresses the points raised during the review process.

We look forward to receiving your revised manuscript.

Kind regards,

Stephen Beckett, Ph.D.

Academic Editor

PLOS Computational Biology

Tobias Bollenbach

Section Editor

PLOS Computational Biology

**Additional Editor Comments:**

Reviewers are enthusiastic about the quality and importance of this work fitting plant disease data with a spatial lattice model. The reviewers raise several suggestions and points of clarification that the authors should consider that may strengthen the manuscript, some of which I summarize below. In particular, one concern shared by reviewers regards explaining the justification for the choice of error threshold: while Table 1 shows the effect of different error thresholds on the number of accepted samples for each class of model, it is unclear why and how the value of 0.025 was selected by the authors. Table 2 and 3 were also a point of contention – I believe this shows the confidence interval around the mean as the range appears to be symmetric about the mean. Reviewers expected to see reported credible intervals in these Tables evaluated directly from Figures 1 and 5. While the confidence and credible intervals have different interpretations and purposes, Figures 1 and 5 suggest the posterior distributions are not necessarily normally distributed, and some may have much wider credible intervals e.g., for beta and beta_F. Such information may hint at how challenging different parameters are to infer, and I would encourage reporting the credible intervals in Tables 2 and 3. Additionally, Reviewer 3 raises an interesting suggestion regarding choice of model-data comparison: might a fitting metric incorporating the spatial aspect of the data improve comparison between the two model classes?

**Journal Requirements:**

**Reviewers' comments:**

Reviewer's Responses to Questions

**Comments to the Authors:**

Reviewer #1: Please see attachment.

Reviewer #2: Best and Cunniffe present epidemiological models for disease spread in a spatial lattice, focusing on plant disease spread in fields or orchards. The topic is enormously important because of the many economically impactful diseases that affect plants throughout the world. Although many crops are certainly not arranged in a lattice, lattices do provide an excellent framework for epidemic modeling with numerous important crops, especially tree crops such as citrus. The authors nicely motivate their investigation and develop different models for the spatio-temporal epidemic process. In particular, they develop a pair approximation differential-equation model for disease spread in a lattice, and consider deterministic and stochastic predictions for a disease of citrus. They compare results with those obtained with a somewhat more traditional dispersal kernel model. I like how they incorporate founder plant infections into their lattice models.

The models, including the parameters, are clearly explained. Others have developed theoretical lattice-based models for epidemics (and see my comment below), but a highly significant aspect of the current work is that the authors took a rigorous approach to fitting their models to actual epidemic data, estimating parameters, and comparing predictions (including the heterogeneity of the predictions). They took a Bayesian approach, and utilized the ABC estimation method since the likelihood is not easily computed. This is appropriate for their complex model. Readers should be able to utilize the authors’ methodology with other models and other biological systems.

Overall, the authors did an excellent job of summarizing the literature on modeling of spatial-lattice models, with one exception. I was surprised that the spatial-lattice epidemic modeling papers of J.A.P. Filipe and colleagues were not mentioned. I realize that there are differences with the current work, such as: Filipe primarily deals with SIS models, but the current manuscript deals with SEI models; Filipe took primarily a theoretical approach, but the current manuscript heavily deals with fitting models to data. Nevertheless, Filipe deals extensively with pair approximations, mean fields, nearest-neighbors, closure, etc. I think the authors of the current work should incorporate how they relate to Filipe’s work. Example papers include (to varying degrees of relevance): Filipe and Gibson (1998; Phil. Trans. Roy. Soc. London B); Filipe and Gibson (2001; Bull. Math. Biol.); Filipe and Maule (2003; Math. Biosciences); and Filipe and Maule (2004; J. Theor. Biol.).

I have just a few specific comments (referenced by line number).

143 It is fine to use an exponential dispersal kernel for this work. However, I think the authors could add some text to the Discussion about the implication of using a fat-tailed dispersal kernel. Empirical studies show about half of observed dispersal gradients have a fat tail.

203 It is not clear to me how the threshold of 0.025 was selected based on the results in Table 1. I think the authors could improve this explanation.

Table 2 (etc.): Bayesian usually refer to “Credible Intervals” and not “Confidence Intervals” for parameters; this better distinguishes between the frequentist interpretation of sampling distributions and the Bayesian interpretation of posterior distributions. There are two main methods for the calculation of the limits of a credible interval, which should apply to bootstraps: Equal-Tailed and Highest Posterior Density. These can differ, especially for skewed posteriors. Maybe I missed it, but I don’t think the authors mentioned the approach they took.

211 Supplemental figure 2 shows a NONLINEAR relation between 1/rho and beta, with close to an asymptotic relation. I think the implications of this could be mentioned.

Fig. 2 legend: Not clear why there is a “Nik” superscript on “model”.

353: Citation 58 gives a year of 2006 (see references). However, here the authors say 2014.

Reviewer #3: See attachement

**Have the authors made all data and (if applicable) computational code underlying the findings in their manuscript fully available?**

Reviewer #1: Yes

Reviewer #2: Yes

Reviewer #3: Yes

PLOS authors have the option to publish the peer review history of their article (what does this mean?). If published, this will include your full peer review and any attached files.

Reviewer #1: No

Reviewer #2: No

Reviewer #3: No

**Figure resubmission:**
---

## [Editor Report · Decision Letter 1]

4 Feb 2026

Dear Dr. Best,

We are pleased to inform you that your manuscript 'Fitting a lattice model with local and global transmission to spread of a plant disease' has been provisionally accepted for publication in PLOS Computational Biology.

Best regards,

Stephen Beckett, Ph.D.

Academic Editor

PLOS Computational Biology

Tobias Bollenbach

Section Editor

PLOS Computational Biology

---

## [Editor Report · Acceptance letter]

PCOMPBIOL-D-25-01589R1

Fitting a lattice model with local and global transmission to spread of a plant disease

Dear Dr Best,

I am pleased to inform you that your manuscript has been formally accepted for publication in PLOS Computational Biology. Your manuscript is now with our production department and you will be notified of the publication date in due course.

With kind regards,

Anita Estes
